# A Zebrafish-Based Platform for High-Throughput Epilepsy Modeling and Drug Screening in F0

**DOI:** 10.3390/ijms25052991

**Published:** 2024-03-04

**Authors:** Sílvia Locubiche, Víctor Ordóñez, Elena Abad, Michele Scotto di Mase, Vincenzo Di Donato, Flavia De Santis

**Affiliations:** 1ZeClinics S.L., Carrer de Laureà Miró, 408-410, 08980 Sant Feliu de Llobregat, Spain; silvia.locubiche@zeclinics.com (S.L.); victor.ordonez@zeclinics.com (V.O.); elena.abad@upf.edu (E.A.); michele.scotto@zeclinics.com (M.S.d.M.); 2Faculty of Medicine and Health Sciences, Institute of Neurosciences, University of Barcelona, 08036 Barcelona, Spain

**Keywords:** zebrafish, CRISPR/Cas9, epilepsy, drug-screening, antiepileptic compounds, neurodevelopmental disorders

## Abstract

The zebrafish model has emerged as a reference tool for phenotypic drug screening. An increasing number of molecules have been brought from bench to bedside thanks to zebrafish-based assays over the last decade. The high homology between the zebrafish and the human genomes facilitates the generation of zebrafish lines carrying loss-of-function mutations in disease-relevant genes; nonetheless, even using this alternative model, the establishment of isogenic mutant lines requires a long generation time and an elevated number of animals. In this study, we developed a zebrafish-based high-throughput platform for the generation of F0 knock-out (KO) models and the screening of neuroactive compounds. We show that the simultaneous inactivation of a reporter gene (*tyrosinase*) and a second gene of interest allows the phenotypic selection of F0 somatic mutants (crispants) carrying the highest rates of mutations in both loci. As a proof of principle, we targeted genes associated with neurodevelopmental disorders and we efficiently generated de facto F0 mutants in seven genes involved in childhood epilepsy. We employed a high-throughput multiparametric behavioral analysis to characterize the response of these KO models to an epileptogenic stimulus, making it possible to employ kinematic parameters to identify seizure-like events. The combination of these co-injection, screening and phenotyping methods allowed us to generate crispants recapitulating epilepsy features and to test the efficacy of compounds already during the first days post fertilization. Since the strategy can be applied to a wide range of indications, this study paves the ground for high-throughput drug discovery and promotes the use of zebrafish in personalized medicine and neurotoxicity assessment.

## 1. Introduction

The zebrafish model has established itself as a reference tool for phenotypic drug screening. Over the last decade, an increasing number of molecules have reached clinical practice upon validation through zebrafish-based assays [1]. To date, several compounds screened through zebrafish-based platforms have been in different phases of clinical trials for a wide array of indications, including cancer, neurological disorders and cardiovascular pathologies [1,2]. The combination of disease modeling through the CRISPR/Cas9 system with different high-throughput phenotyping platforms have greatly contributed to shortening the time of preclinical compound testing in zebrafish larvae [3]. So far, most of the zebrafish-based drug discovery pipelines have been based on the use of isogenic lines carrying a mutation in the zebrafish orthologue of a human disease-associated gene [4,5].

Nevertheless, isogenic lines require a long generation time while, to date, novel genetic targets have been identified at a fast pace thanks to large-scale patient DNA sequencing studies [6]. Thus, high-throughput methods for the rapid generation of animal disease models are required in order to obtain, in a timely manner, preliminary data on the efficacy of compounds modulating the novel targets. Indeed, several strategies have been optimized to shorten the time required to (i) generate zebrafish genetic models, (ii) characterize their loss-of-function phenotype and (iii) perform target validation or drug screenings. The refinement of high-throughput genotyping approaches combined with the development of multiplex gene targeting in zebrafish embryos allow the assessment of the effect of a gene loss of function as early as in the F0 generation [7,8,9].The achievement of high rates of CRISPR/Cas9-mediated double-strand breaks in micro-injected zebrafish embryos promotes biallelic gene inactivation and, as a consequence, the generation of de facto mutants (defined as crispants) [8]. In fact, recent reports have validated the use of crispants in target prioritization and disease modeling in cardiovascular indications [10]. The crispants carrying the highest rate of loss-of-function mutations will behave as homozygous mutants, allowing the direct identification and analysis of mutant phenotypes, thus reducing the time and costs required to reach homozygous in the F2 generation [11]. Importantly, it has been also shown that continuous traits, with phenotypes varying within a continuous range, can be assessed in crispants, despite the diversity of null alleles in the F0 generation, and that multiple genes can be targeted at the same time in the same individual [12].

Nonetheless, although this kind of approach greatly reduces the time needed for functional genomic studies, it is still associated with a very high intra- and inter-individual variability. For this reason, to establish causality between the induced mutation and the observed phenotype, time-consuming individual genotyping has to be performed after injected larvae are analyzed.

The study of neurodevelopmental disorders (NDDs) is one of the research fields that might benefit more from the establishment of a robust pipeline allowing the assessment of gene function in F0 larvae. It has been estimated that NDDs affect up to 15% of children and adolescents worldwide and cause deficits in mental performance, adaptive behavior and motor skills [13]. The genetic causes underlying these disorders are extremely complex, since, although common molecular players have been identified [14], the nature of the disease-causing mutations is heterogeneous, resulting in variable clinical outcomes [15].

This is particularly true for epilepsy, which is regarded as an NDD due to the contribution of multiple developmental variables, including congenital brain abnormalities and aberrant neuronal signaling throughout embryonic life [16]. Epilepsy is one of the most common neurological disorders, with over 70 million people diagnosed worldwide [17,18]. This disorder is characterized by the appearance of seizures, a consequence of an imbalance between excitatory and inhibitory circuits, causing both cognitive and psychological impairment and increasing the risk of early death [18]. The etiology underlying this disease is variable, genetic mutations being one of the most important causes, resulting in defects in ion channels located in the neuronal membrane or alterations in other neuronal mechanisms [19]. Many pediatric epilepsy patients present frequent seizures, events that damage and alter their quality of life. Seizures can be provoked by various sensory inputs and when they are produced in response to light, color, or patterns, it is considered photosensitive epilepsy. Photosensitivity occurs in several epileptic syndromes, being particularly prevalent in genetic generalized epilepsies [20]. Moreover, several epilepsies are resistant to the treatments currently available, demonstrating the ongoing need to test novel anti-seizure medications (ASM). In the search for potentially effective ASM, several in vitro assays, involving cell culture, brain slice culture, organoids, or primary cultures of neurons, have been developed [21]. These approaches can be used for a detailed analysis of the effect of a given ASM at the cellular level. However, the use of these models makes it challenging to evaluate the toxicity and efficacy of compounds, since mechanisms such as drug metabolism or penetration through the blood–brain barrier cannot be readily assessed, thus limiting the obtained information. These considerations highlight the need to complement data obtained in vitro with in vivo models to perform a comprehensive evaluation of novel potentially therapeutic drugs under conditions closer to a whole-organism scenario. As such, in vivo evaluation systems have been extensively described, including the zebrafish [22].

Different structures of the central nervous system (CNS) of zebrafish, such as the telencephalon, the cerebellum, the spinal cord and the medulla oblongata are homologous to those in mammalian brains. In addition, the properties of neurons and neural circuits are conserved between zebrafish and mammals [23]. There are several reports on the use of zebrafish in epilepsy research based on drug-induced and genetically-induced seizures [24]. The gold standard pharmacological model relies on the administration of pentylenetetrazole (PTZ), a typical convulsant [25,26], determining an increase in locomotor activity and the appearance of a characteristic circular swimming pattern. Of all animal models for seizure and epilepsy, pentylenetetrazole-induced seizures are classified as a model for generalized seizures, as opposed to partial or focal seizures. As for the genetic model, the most widespread is a mutant of the *scn1lab* gene characterized by spontaneous seizures and altered behavior in zebrafish larvae that can be recorded by electrophysiology and are comparable to epilepsy in humans. These models have proven valuable for the validation of commonly used ASMs and the repurposing of novel compounds [4,27,28].

Taking into account the advantages that the zebrafish model provides and given the necessity of expanding the toolbox for modeling epilepsy, in order to streamline the discovery of novel ASMs, in this study we aimed for the establishment of a workflow allowing scientists to use somatic knock-out larvae to analyze the function of genes involved in the pathogenesis of childhood NDD. Our objective is to develop a zebrafish-based comprehensive platform enabling the functional validation of common and de novo rare loss-of-function mutations in a high-throughput fashion. To this end, we propose a strategy based on the generation of crispants, co-targeting the gene of interest with a reporter gene whose loss of function is associated with an easily detectable phenotype, the *tyrosinase* gene. Moreover, we coupled highly efficient gene inactivation with the automated analysis of the morphological developmental defects potentially induced by the gene inactivation and with a complex multiparametric behavioral analysis to describe seizure-like events. As proof of principle, we knocked out a group of six genes associated with childhood epilepsy, characterized their loss-of-function phenotype and proved their response to a selection of ASMs.

## 2. Results

### 2.1. Tyrosinase Loss of Function Allows the Selection of Crispants Carrying High Rate of Mutations in Targeted Genes

To reduce variability and maximize the utility of an F0 knock-out approach, it is necessary to cluster together genetically homogeneous populations. To this end, distinguishing between larvae carrying high and low rates of mutations is needed. To be able to perform such selection, we hypothesized that, if two genes are targeted simultaneously, their sequences will undergo a comparable mutagenesis process and, as a consequence, the presence of a phenotype associated with the loss of function of one gene could be employed as a reporter for the efficient inactivation of the second gene. The rationale is that the efficiency of double-strand breaks (DSB) is largely related to the injection procedure and timing. As such, the embryo stage at the time of the injection as well as injection accuracy are critical to achieving high DSB rates. Therefore, a complete loss of function of the reporter gene might suggest correct injection timing and high injection accuracy. To this end, we thought to target, together with the candidate gene of interest, the *tyrosinase* (*tyr*) locus. The *tyr* gene encodes a protein involved in melanin production and its disruption results in the absence of pigmentation, an easily identifiable phenotype that could be employed to select larvae in which the CRISPR/Cas9 system has efficiently disrupted the target genomic sequences [29].

First, we wanted to prove that *tyr* inactivation does not cause side developmental phenotypes other than the expected pigmentation deficiency in zebrafish larvae. To exclude the occurrence of morphological developmental defects in *tyr* crispants, wild-type one-cell stage embryos were injected with a mix containing Cas9/scrambled-sgRNA or Cas9/*tyr*-sgRNA. As expected, starting from 48 hpf, embryos carrying biallelic mutations in the *tyr* gene could be identified by the absence of pigmentation. This difference became particularly clear at 120 hpf (Figure 1A). At this stage, we performed a comprehensive characterization of a panel of 9 qualitative (Table 1) and 3 quantitative (body length, eye diameter and heart area) phenotypes [30]. Importantly, we did not detect any significant morphological differences between scrambled and *tyr* crispants for qualitative phenotypes. A very mild but significant alteration was observed in the body length and eye diameter of *tyrosinase* crispants (mean ± SEM: Body length: scrambled = 3693 μm ± 12.1, n = 75; *tyr* = 3608 μm ± 16.51, n = 68. Eyes diameter: scrambled = 345 μm ± 1.6, n = 75; *tyr* = 328 μm ± 1.5, n = 68). In contrast, no differences were observed in the area of the heart chamber (mean ± SEM: scrambled = 27,880 μm^2^ ± 640.4, n = 75; *tyr* = 29,436 μm^2^ ± 645.7, n = 68) (Figure 1A–D, Table 1).

Having assessed the general morphology, we verified that larval locomotion and behavior were not affected by *tyr* loss of function. To this end, we analyzed the motor activity of crispants in response to different stimuli. We re-injected both scrambled and *tyr*-specific sgRNAs to track 120 hpf control and *tyr* crispants while they were exposed to different stimuli. In the first phase of the trial, larvae were exposed to alternating dark/light cycles. It has been observed that zebrafish have a stereotyped behavior in which they tend to increase their locomotion during the dark phases and rest during the light period [31,32]. Therefore, this step of the trial allowed us to evaluate larval locomotion (total distance moved (mm)), spotting potential motor defects. The second half of the trial was focused on identifying light-induced seizures. Flashlight stimulus has been previously described as a seizure-inducing stimulus, showing an improvement in the detection of convulsant behaviors when compared with other paradigms [24]. The manifestation of convulsive seizures is evaluated as the happening of erratic movement and quantified by an increased swimming velocity and a heightened number of changes in the trajectory. In both parts of the test, *tyr* crispants did not show any significant difference when compared to control crispants (mean ± SEM: Total distance moved: scrambled = 1511 mm ± 57.30, n = 126; *tyr* = 1399 mm ± 63.75, n = 122. Maximum velocity: scrambled = 10.83 mm/s ± 0.42, n = 127; *tyr* = 10.29 mm/s ± 0.4, n = 121. Number of turns: scrambled = 1.1 ± 0.12, n= 117; *tyr* = 1.4 ± 0.13, n = 120) (Figure 1E–G). Therefore, our data confirm that *tyr* loss of function does not cause motor defects and does not induce epileptic seizures.

Having verified that *tyr* disruption does not affect larval morphology or behavior, we wanted to prove that the loss of pigmentation could be employed as a reporter for the efficient disruption of the coding sequence (CDS) of a second gene. We tested our approach on the zebrafish orthologues of six genes whose loss of function had been associated with different kinds of genetic epilepsy (Table 2). One of the epilepsy-associated genes (KCNQ2) has two orthologues in zebrafish (*kcnq2a* and *kcnq2b*). In this case, we generated three types of crispants: two single crispants where only one of the two paralogues was targeted (single crispants) and a double crispant in which both paralogues were simultaneously targeted.

To confirm our hypothesis, we injected one-cell stage wild-type embryos with a mix containing the Cas9 protein together with three sgRNAs (one targeting the *tyr* and two targeting one of the selected target genes). To verify that the mutation rates of the two targeted genes correlated, we extracted the genomic DNA from individual pigmented and unpigmented larvae and sequenced the second locus targeted (Appendix A). As expected, un-pigmented larvae displayed a higher mutagenesis rate than the one observed in their pigmented siblings (mean ± SEM: *adgrg1* not pigmented = 88.85 ± 4.4, n = 20; *adgrg1* pigmented = 28.6 ± 12.14, n = 12; *gabra1* not pigmented = 95 ± 5, n = 20; *gabra1* pigmented = 3.7 ± 3.2, n = 12; *kcnq2a* (single crispants) not pigmented = 94.24 ± 4.5, n = 21; *kcnq2a* (single crispants) pigmented = 0 ± 0, n = 12; *kcnq2a* (double crispants) not pigmented = 91.74 ± 2.38, n = 21; *kcnq2a* (double crispants) pigmented = 0.538 ± 0.4, n = 13; *kcnq2b* (single crispants) not pigmented = 71.05 ± 8.93, n = 20; *kcnq2b* (single crispants) pigmented = 25 ± 13.056, n = 12; *kcnq2b* (double crispants) not pigmented = 73.11 ± 8.05, n = 19; *kcnq2b* (double crispants) pigmented = 0 ± 0, n = 12; *pcdh19* not pigmented = 78.25 ± 6.5, n = 20; *pcdh19* pigmented = 33.15 ± 12.5, n = 13; *scn1lab* not pigmented = 98.17 ± 0.9, n = 29; *scn1lab* pigmented = 91 ± 5.7, n = 11; *ube3a* not pigmented = 86.8 ± 6.18, n = 20; *ube3a* pigmented = 8.3 ± 8.3, n = 12) (Figure 1H). As previously described [29], the efficiency of the CRISPR/Cas9 machinery is locus-dependent, and we observed that different loci had a different susceptibility to the Cas9 endonuclease activity. Nevertheless, in all genes analyzed, more than 60% of unpigmented larvae displayed a mutation rate higher than 75% (Appendix A), leading us to the conclusion that it is possible to use *tyr* loss of function as a reliable reporter for Cas9 cutting efficiency.

### 2.2. adgrg1, gabra1, pcdh19, scn1lab and ube3a Crispants Show Epilepsy-like Behavior

Having confirmed that the loss of pigmentation represents a powerful screening method to identify gene-specific crispants that carry a high rate of mutations, we used this approach in the following experiments. For the study of the phenotype induced by the loss of function of the different candidate epilepsy genes, we injected one-cell stage wild-type embryos with Cas9/sgRNA complexes, one sgRNA targeting the *tyr* locus and two sgRNA targeting the epilepsy gene of interest. For the negative control, we co-injected the *tyr* sgRNA and a scrambled sgRNA. We selected the unpigmented larvae for the experiments.

To evaluate the effects of the loss of function of the selected genes, we first looked for the presence of morphological alterations in the different gene-specific crispants. We observed that *adgrg1* crispants displayed significant morphological phenotypes (reduced size and presence of body curvature) (Appendix A), possibly due to a developmental delay or to hypotonia and muscular defects related to the generated loss of function. The inactivation of other epilepsy-associated genes did not result in significant morphological deficiencies, with the exception of *gabra1* and *pcdh19* crispants (minor but significant decrease in body length) and *kcnq2a* and *kcnq2b* crispants (minor but significant increase in body length and eye diameter) (Appendix A).

We next analyzed the presence of locomotion alterations or the occurrence of epileptic-like phenotypes.

First, we analyzed the locomotion activity of control and gene-specific crispants in response to alternating dark/light cycles. Interestingly, most crispants did not show differences in their motor behavior in response to the dark/light changes (total distance moved, mean ± SEM: *adgrg1*: scrambled = 1728 mm ± 135.2, n = 90; crispants = 1810 mm ± 110.2, n = 93. *kcnq2ab*: scrambled = 1106.31 mm ± 63.03, n = 76; *kcnq2a* crispants = 1328.29 mm ± 66.85, n = 75; *kcnq2b* crispants = 1184.28 mm ± 54.86, n = 77; *kcnq2ab* crispants = 1143.86 mm ± 69.67, n = 73. *pcdh19*: scrambled = 1299.51 mm ± 84.04, n = 60; crispants = 1319.76 mm ± 88.47, n = 53. *ube3a*: scrambled = 1062.03 mm ± 106.53, n = 47; crispants = 1041.61 mm ± 85.23, n = 44), (Figure 2A, Figure 3A,E and Figure 4E)). In contrast, *scn1lab* and *gabra1* crispants showed reduced locomotion in comparison to scrambled crispants (total distance moved, mean ± SEM: *scn1lab*: scrambled = 1062.03 mm ± 106.53, n = 47; crispants = 750.73 mm ± 64.19, n = 44; *gabra1*: scrambled = 1299.51 mm ± 84.04, n = 60; crispants = 894.33 mm ± 86.63, n = 56), while *kcnq2a* displayed a moderate but significantly increased movement (total distance moved, mean ± SEM: scrambled = 1106.31 mm ± 63.03, n = 76; *kcnq2a* crispants = 1328.29 mm ± 66.85, n = 75)) (Figure 2E and Figure 4A).

It has been previously demonstrated that the incubation of zebrafish larvae with PTZ causes seizures associated with a strong behavioral response that can be quantified by analyzing kinematic parameters such as the total distance moved or the maximum velocity [17]. In our experiment, crispants were exposed to two sub-optimal concentrations of PTZ (1 and 3 mM) [25,42], and their locomotion activity was monitored for a sustained light period (15 min). The presence of locomotion alterations and spontaneous convulsions was evaluated by analyzing the maximum velocity (mm/s) reached by each larva during the trial.

Interestingly, we found that *pcdh19* and *ube3a* had an increased sensitivity to PTZ. Indeed, if the two genes were inactivated, the behavioral response of larvae treated with PTZ 3 mM was significantly stronger than the one observed in scrambled controls exposed to the same concentration of the compound (Maximum velocity: mean ± SEM: scrambled = 53.71 mm/s ± 1.68, n = 26; *pcdh19:* crispants = 87.78 mm/s ± 13.09, n = 24; *ube3a:* crispants = 72.50 mm/s ± 7.39, n = 27. Figure 3F and Figure 4F). In both cases, the incubation with a lower concentration of PTZ (1 mM) was not sufficient to induce seizure-like activity (Maximum velocity: mean ± SEM: scrambled = 37.76 mm/s ± 2.14, n = 32; *pcdh19:* crispants = 30.84 mm/s ± 4.71, n = 29; *ube3a:* crispants = 33.21 mm/s ± 5.17, n = 27). Differently, the KO of the other epilepsy-associated genes did not increase the susceptibility of crispants to PTZ (Figure 2B,F, Figure 3B and Figure 4B) (PTZ 1 mM: Maximum velocity: mean ± SEM: *adgrg1*: scrambled = 37.77 mm/s ± 2.14, n = 32; crispants = 31.64 mm/s ± 4.19, n = 31. *gabra1*: scrambled = 35.95 mm/s ± 2.63, n = 30; crispants = 32.56 mm/s ± 4.37, n = 43; *kcnq2a*: scrambled = 24.59 mm/s ± 3.62 n = 32; crispants = 22.23 mm/s ± 3.71, n = 31. *kcnq2b:* scrambled = 24.59 mm/s ± 3.62, n = 32; crispants = 33.49 mm/s ± 3.19, n = 31. *kcnq2ab:* scrambled = 24.59 mm/s ± 3.62, n = 32; crispants = 30.59 mm/s ± 3.81, n = 30. *scn1lab:* scrambled = 35.95 mm/s ± 2.63, n = 30; crispants = 41.84 mm/s ± 2.73, n = 45; PTZ 3 mM: Maximum velocity: mean ± SEM: *adgrg1*: scrambled = 53.71 mm/s ± 1.67, n = 26; crispants = 53.38 mm/s ± 1.43, n = 28. *gabra1*: scrambled = 51.24 mm/s ± 1.69, n = 22; crispants = 50.62 mm/s ± 5.33, n = 40; *kcnq2a*: scrambled = 38.01 mm/s ± 5.37, n = 31; crispants = 45.44 mm/s ± 3.25, n = 27. *kcnq2b:* scrambled = 38.01 mm/s ± 5.37, n = 31; crispants = 40.83 mm/s ± 5.48, n = 29. *kcnq2ab:* scrambled = 38.01 mm/s ± 5.37, n = 31; crispants = 43.39 mm/s ± 5.56, n = 30. *scn1lab:* scrambled = 51.24 mm/s ± 1.69, n = 22; crispants = 49.40 mm/s ± 1.35, n = 36).

When exposed to intermittent light as an epileptogenic stimulus, different crispants showed a different response. In particular, the disruption of one or both paralogues of the *kcnq2* gene did not cause photosensitive seizures, as confirmed by the absence of a significant increase of the motor activity in response to the light stimuli (maximum velocity, mean ± SEM: scrambled = 11.29 mm/s ± 0.73, n = 59; *kcnq2a* = 12.71 mm/s ± 0.67, n = 49; *kcnq2b* = 12.74 mm/s ± 0.82, n = 58; *kcnq2ab* = 11.87 mm/s ± 0.85, n = 57; number of turns, mean ± SEM: scrambled = 1.21 ± 0.20, n = 52; *kcnq2a* = 1.47 ± 0.22, n = 50; *kcnq2b* = 1.71 ± 0.22, n = 59; *kcnq2ab* = 1.89 ± 0.22, n = 57) (Figure 3C,D). Similarly, the inactivation of *ube3a* only caused a moderate increase in the number of turns performed by crispants exposed to light flashes (maximum velocity, mean ± SEM: scrambled: 11.72 mm/s ± 1, n = 45; *ube3a* = 13.16 mm/s ± 1.05, n = 47; number of turns, mean ± SEM: scrambled = 1.02 ± 0.20, n = 42; *ube3a* = 2.08 ± 0.36, n = 46) (Figure 4G,H). In contrast, the exposure to quick and repeated light flashes had a strong impact on the behavior of *adgrg1*, *gabra1*, *pcdh19* and *scn1lab* crispants. The presence of seizure-like behaviors could be visually detected by analyzing the trajectory followed by each larva in the two seconds after the exposure to the light stimulus. While crispants injected with scrambled sgRNAs followed a linear trajectory along the wall of the well, *adgrg1*, *gabra1*, *pcdh19* and *scn1lab* crispants moved along an erratic trajectory in which they crossed the well and changed direction multiple times (Figure 2C,G, Figure 3G and Figure 4C). The analysis of two kinematic parameters (the maximum velocity reached in the two seconds following the exposure to the flash and the number of turns realized in the same time frame) confirmed this observation (Figure 2D,H, Figure 3H and Figure 4C). Both parameters are significantly increased if compared to scrambled control (Maximum velocity: mean ± SEM: *scn1lab*: scrambled = 11.73 mm/s ± 0.99, n = 45; crispants = 17.95 mm/s ± 1.47, n = 42. *pcdh19*: scrambled = 10.09 mm/s ± 0.79, n = 58; crispants = 12.91 mm/s ± 1.10, n = 57. *gabra1*: scrambled = 10.09 mm/s ± 0.79, n = 58; crispants = 21.56 mm/s ± 1.55, n = 58. *adgrg1*: scrambled = 6.74 mm/s ± 0.59, n = 91; crispants = 10.6 mm/s ± 0.59, n = 87. Number of turns: mean ± SEM. *scn1lab*: scrambled = 1.02 ± 0.20, n = 42; crispants = 3.38 ± 0.46, n = 40; *pcdh19*: scrambled = 0.69 ± 0.14, n = 51; crispants = 2.09 ± 0.34, n = 59; *gabra1*: scrambled = 0.69 ± 0.14, n = 51; crispants = 5.21 ± 0.56, n = 59; *adgrg1*: scrambled = 0.34 ± 0.07, n = 72; crispants = 2.46 ± 0.24, n = 91), suggesting that the mutation of *adgrg1*, *gabra1*, *pcdh19* and *scn1lab* CDS increases the susceptibility to light-induced seizures.

### 2.3. Multiparametric Analysis of Behavioral Response to Light Flashes Allows Fine Characterization of Photosensitive Epilepsy

To detect spontaneous seizures (e.g., in response to incubation with PTZ), the behavior of larvae has to be recorded for a relatively long time window. Indeed, it is not possible to predict when the epileptic event is going to occur, and, to increase the probability of tracking a seizure, it is necessary to monitor the locomotion activity of larvae for a long period of time. In contrast, the use of flashes of light as a causative convulsive stimulus allows to temporally control the manifestation of seizure-like events, making it possible to assess and characterize behavioral alterations in a time-convenient manner.

Bearing in mind these considerations, we decided to focus on light-induced seizures in the following part of this study.

Epilepsy is characterized by a high heterogeneity including differences in severity, symptoms and causes, with structural, genetic, metabolic, autoimmune and infection-related causes. There are different manifestations of epilepsy, leading to a complex classification of the seizures (e.g., focal or generalized and motor or non-motor) [43,44]. Moreover, comorbidities are very relevant and conditions such as depression, anxiety and dementia are more common in people with epilepsy [45]. Therefore, to better evaluate epilepsy in whole organisms and characterize the possible effects of different therapies, a comprehensive phenotype evaluation is required.

Taking this information into account, we decided to perform a more detailed analysis of the behavioral response of zebrafish larvae to an epileptogenic stimulus in order to better classify the observed epileptic-like responses.

To extract the most possible information from our dataset, we built a principal component analysis (PCA), considering those kinematic variables having the greatest relation and relevance in the study of epileptic seizures. To perform such analysis, we focused on parameters associated with a modification of the larval position or orientation in the space (Maximum velocity (mm/s), number of angle turns, maximum acceleration (mm/s^2^), angular velocity (deg/s)) and variables reflecting a movement alteration (e.g., tremors or freezing behaviors) that do not cause a change in the position or orientation of larvae (mobility in the arena (%) and cumulative duration (s) of three different mobility states, immobile, mobile and highly mobile) (Figure 5A and Appendix A).

Data from all replicates and all different crispants were pooled in this multivariate analysis. Interestingly, we noticed that the samples were distributed in the PCA plot along different levels of overall “activity”. Indeed, variables associated with increased motor activity (e.g., maximum velocity, maximum acceleration, highly mobile state) pointed toward a similar direction, one opposed to that of variables reflecting low motor activity (e.g., immobile state) (Figure 5A). To our surprise, we did not observe a clear clusterization of crispants versus scrambled controls but a dispersed sample distribution in which most observations localized in the center of the plot.

Nonetheless, we noticed that a number of samples showed a highly aberrant behavior, with higher values in different kinematic parameters (Figure 5B). We reasoned that analyzed larvae could be classified in different groups depending on their activity level and we speculated that the crispants of genes having a positive association with photosensitive epilepsy would have an increased representation in the more active group.

To explore this possibility, we decided to measure the Mahalanobis distance (MD) of all the observations in order to evaluate how each flash response differs from the average of the entire population. We speculated that this analysis would allow us to define the behavioral fingerprint of each larva and to classify their response to epileptogenic flashes of light based on the intensity of the observed motor activity.

Indeed, we distinguished and classified two regions of activity in our population: a region of low activity (MD < 2.5 and *p* value > 0.1) and a region of high activity (MD > 2.5 and *p* value < 0.1) (Figure 5C,D). This observation suggests that larvae with increased photosensitivity are prone to display a highly aberrant behavior in response to exposure to flashes of light, constituting a population of outliers distinguished by their higher motor activity. We assume that these extreme behaviors correspond to epileptic-like seizures characterized by aberrant and excessive movements.

For each experimental group, we calculated the proportion of larvae whose level of activity was statistically classified as “low” or “high” (Table 3).

Interestingly, we observed that four crispants (*adgrg1*, *gabra1*, *pcdh19* and *scn1lab*) displayed an increased proportion of seizure-like responses when compared to control larvae. Among these, the crispants with the highest proportion of “highly active” responses were *gabra1*, accounting for a percentage of hyperactive responses double the one observed in scrambled controls, and *scn1lab*, showing four times more “active responses” than scrambled controls (Figure 5E), suggesting that the inactivation of these genes had the strongest association with photosensitivity and, as a consequence, promoted the manifestation of seizure-like events in response to flashes of light.

### 2.4. The Incubation of scn1lab Crispants with Antiepileptic Compounds Protects against Seizure-like Events

Having proved that *scn1lab* crispants display the highest susceptibility to light-induced seizures, we wanted to evaluate the effect of antiepileptic compounds on the behavioral phenotype described above.

Epilepsies, and especially childhood epilepsies, are known to present pharmaco-resistant seizures, highlighting the need of designing targeted therapeutic strategies. With the objective of evaluating different known antiepileptic drugs, three different compounds were selected: valproic acid, which enhances the inhibitory neurotransmission through GABA potentiation and sodium and calcium voltage-gated channels modulation, overall decreasing neuronal excitability and firing rate [46,47,48]; topiramate, whose mechanism of action is the increase of GABA activity and inhibition of glutamatergic activity (NMDAR antagonism) through GABA potentiation and sodium and calcium channels modulation [49]; and fenfluramine, which acts as an agonist of the serotonin 5-HT2 receptors and σ1 receptor positive modulator [50,51,52]. We speculated that the treatment of photosensitive crispants with effective antiepileptic drugs would reduce the number of seizure-like events, resulting in a reduced presence of treated crispants in the region of “high activity”. To test this hypothesis, we incubated *scn1lab* crispants with two concentrations of the aforementioned drugs.

Upon incubation with the test compounds, control and gene-specific crispants were exposed to flashes of light to induce a convulsive behavior. First, we evaluated the efficacy of the tested ASMs from a qualitative point of view, examining the trajectory followed by larvae in response to the flashes of light. As expected, the trajectory of DMSO-treated *scn1lab* crispants appeared more complex and fragmented than that of DMSO-treated scrambled control (Figure 6A). In contrast, treated larvae followed a less complex trajectory, even if we observed differences among different drugs (Figure 6A). Indeed, *scn1lab* crispants treated with the two selected concentrations (17.5 µM and 35 µM) of fenfluramine and with the highest concentration of valproic acid (100 µM) followed a linear trajectory comparable to that of control individuals. In contrast, larvae treated with the lowest concentration of valproic acid (50 µM) and with the two chosen concentrations of topiramate (50 µM and 100 µM) displayed a twistier trajectory (less complex than the one followed by DMSO-treated *scn1lab* crispants but more complex than the one described in scrambled control crispants).

Then, the kinematic parameters of each larva were extracted and analyzed, and the response of each sample was classified into the “low” (normal behavior) or “high” active region (epileptic-like behavior).

As expected, for each of the treatments, the most represented group in the “high” active region was that of DMSO-treated crispants (Figure 6B–D). These data confirm that *scn1lab* inactivation triggers an increased susceptibility to epileptic seizures. In contrast, DMSO-treated scrambles constituted only a minority of the observations in the higher active regions (Figure 6B–D). Statistically, the proportions of *scn1lab* crispants in the “active” region were significantly different compared to scrambled larvae (binomial test with *p*-value < 0.05) in each of the three drug experiments. Most importantly, the treatment with both concentrations of fenfluramine appeared to protect against the manifestation of aberrant light-induced behavior (Percentage of highly active flash responses: scrambled DMSO = 5.7%, n = 401 responses; *scn1lab* DMSO = 12.2%, n = 491 responses; fenfluramine 17.5 µM = 3.7%, n = 54 responses; fenfluramine 35 µM = 4.1, n = 49 responses). The treatment with topiramate did not show a protective effect (Percentage of highly active flash responses: scrambled DMSO = 3.7%, n = 402 responses; *scn1lab* DMSO = 7.1%, n = 493 responses; topiramate 50 µM = 5.5%, n = 109 responses; topiramate 100 µM = 7.3, n = 124 responses). In the case of valproic acid, the lowest concentration did not protect against epileptic-like behaviors while the highest concentration efficiently protected against light-induced aberrant movement (Percentage of highly active flash responses: scrambled DMSO = 5.8%, n = 400 responses; *scn1lab* DMSO = 12.2%, n = 490 responses; valproic acid 50 µM = 9.6%, n = 178 responses; valproic acid 100 µM = 6, n = 182 responses).

Taken together, our data demonstrate that F0 *scn1lab* mutants can be employed as a useful tool in the screening of anti-epileptic compounds.

## 3. Discussion

### 3.1. An Optimized Workflow for Genetic Target Validation in Zebrafish F0 Larvae

In this study, we successfully developed an effective procedure to maximize the accuracy and the translatability of crispants-based target validation approaches for NDD preclinical studies. NDDs are a group of extremely diverse disorders with a complex etiology and a broad symptomatic spectrum, often caused by the inheritance of genetic risk factors [53]. In the past years, a number of point mutations, deletions or chromosomal alterations have been confirmed to be causative of different forms of genetic epilepsy, an NDD in which frequent seizures strongly impair the quality of life of affected patients [54].

As sequencing technologies have continued to advance over the past few years, an increasing number of new variants have been found by multinational collaborations that analyzed large cohorts of epilepsy patients [55,56]. Global efforts such as the Epi25 collaborative or the International League Against Epilepsy (ILAE) Consortium on Complex Epilepsies are generating and analyzing sequences of patient cohorts at a much faster pace than the aforementioned methods allow to test for causality and screen for novel ASM. This is also the case for other NDDs [15].

Taking these considerations into account, it is essential to have a tool that allows testing the new variants in a timely and sustainable manner. So far, the use of zebrafish F0 mutants has allowed scientists to mimic human genetic diseases in a fast, cost-effective and ethically sound manner, offering great opportunities in preclinical research [3,57]. Nonetheless, this experimental approach is associated with a relatively high variability of mutation rates and, to strictly correlate the inactivation of a candidate gene with an observed phenotype, it is necessary to analyze the percentage of mutations carried by each screened larva. Given the fast integration of the CRISPR/Cas9 system into zebrafish laboratories over the last years and the common opinion that high efficiency of double-strand breaks is often obtained, this genotype/phenotype correlation is not routinely performed. To overcome this limitation, we have demonstrated that it is possible to visually identify crispants carrying the highest rate of mutations in a gene of interest by co-targeting the *tyr* CDS. Indeed, *tyr* disruption induces pigmentations defects, a phenotype that can be employed as a reporter of Cas9 activity. We have shown that a simple protocol optimization—the addition of a sgRNA targeting a “reporter” gene—allows to enhance robustness and reduce the time required to carry on F0-based approaches. Our strategy allows the preselection of de facto mutant larvae before performing phenotypic characterization, thus enabling the study of less genetically heterogeneous populations for inferring gene function. We hypothesize that a similar result can be obtained by targeting other sequences whose inactivation results in a visually detectable phenotype (e.g., the GFP CDS in a transgenic reporter line), making this approach extremely flexible and versatile.

Although through this approach the genetic heterogeneity of the larvae is reduced before functional characterization, there are two limitations that need to be acknowledged. The first is that some larvae of the selected unpigmented pools might carry a low rate of mutations, thus not displaying a loss-of-function phenotype. We have shown that the percentage of these larvae is usually low (Appendix A) and thus it should not lead to the appearance of false negatives in the analysis. Additionally, the mutations that are induced by the CRISPR/Cas9 system are random and not all of them disrupt the gene open reading frame; thus, a variable number of cells would not be knocked out for the gene of interest. In this case, we suggest a deletion strategy that maximizes the possibilities of inducing loss of function, either by removing the translational start site from the coding region or by targeting entire exon sequences that encode essential functional protein domains.

Another limitation of the strategy is that only loss-of-function mutations can be induced. Nevertheless, the advent and implementation of new techniques such as base editing [58] and prime editing [59] allow the introduction of patient-specific mutations into the zebrafish genome, opening the possibility of high-throughput personalized screenings. Along this line, Rossello et al. have recently shown high efficiency in F0 disease modeling with base editing co-targeting genes of interest and *tyr* loci [60].

### 3.2. Addressing the Phenotypic Heterogeneity of Epilepsy Combining Morphological and Behavioral Assays

A characteristic of NDD is the high phenotypic heterogeneity detected in human patients. For example, in the case of epilepsy, there is a broad spectrum of phenotypes, with many different seizure types and leading to various conditions such as epileptic encephalopathies, in which epileptiform abnormalities contribute to progressive brain dysfunction; this disorder can be associated with different comorbidities, including mental retardation and developmental delays [61,62], making it challenging to cover these variable phenotypes with a defined experimental assay. Taking into account this phenotypic heterogeneity, we combined the analysis of the morphological and behavioral phenotypes induced by the loss of function of six genes associated with different forms of childhood genetic epilepsy.

The fine-tuned morphological evaluation performed in this study allowed us to observe developmental defects (reduced size and body curvature) in *adgrg1* crispants. These features mirror the developmental delay and muscular weakness observed in some patients affected by polymicrogyria, the pathological condition caused by ADGRG1 mutations in humans [33].

To investigate the presence of neurodevelopmental defects and to verify the occurrence of epileptic-like seizures, we have used behavioral phenotyping of zebrafish larvae to describe larval basal locomotion and to model seizure-induced changes in larval movement. 

Our analysis revealed that the basal locomotion of *gabra1* and *scn1lab* crispants is reduced compared to that of the control scrambled. These data suggest that these crispants might be affected by a neurodevelopmental delay that impairs motor coordination. Interestingly, motor deficits have been described in *scn1lab* stable mutants [63] and are observed in Dravet syndrome patients, suggesting that our genetic manipulation correctly mimics human disease [64]. 

In order to account for the phenotypic variability of epilepsy, we have employed two epileptogenic stimuli: PTZ as a chemical inductor of seizures and flashes of light to detect photosensitive epilepsy. 

Interestingly, we found that the inactivation of most target genes caused an increased susceptibility to one or both triggers. The loss of function of *adgrg1*, *gabra1* and *scn1lab* resulted in increased photosensitivity but did not enhance the epileptogenic activity of PTZ. The photosensitivity induced by the loss of function of *scn1lab* [65] and *gabra1* [66] has been already described, confirming that our somatic KO approach consistently replicates phenotypes already observed in stable mutant lines. Most importantly, the increased sensitivity of *adgrg1* to flashing lights represents a novelty described in this work for the first time. 

The inactivation of *pcdh19* and *ube3a* increased the sensitivity to PTZ. *Pcdh19* crispants also showed an increased response to flashing light while *ube3a* inactivation only had a minor impact on light-induced seizures. Interestingly, *pcdh19* has been previously associated with neuronal hyperactivity in zebrafish [67], but, to our knowledge, this study demonstrated for the first time its association with seizure-like behavior.

The targeting of the two orthologues of KCNQ2 did not cause PTZ- or light-induced seizures. As mentioned above, epileptic disorders have different symptomatic manifestations and epileptic seizures can be triggered by different stimuli (e.g., fever) or be characterized by an increase in neural circuit excitation that does not converge in the occurrence of aberrant movement. Our current analysis is based on the relatively straightforward automated tracking of zebrafish larvae in response to the given stimuli. The flexibility of the behavioral setup employed in this study enabled us to test different triggers to induce epilepsy (e.g., increased temperature) or to evaluate spontaneous seizures over a longer observation time, allowing us to potentially cover a broader spectrum of epileptic disorders. 

### 3.3. The Characterization of Photosensitive Epilepsy with In-Depth Analysis of Larval Behavior

To better characterize photosensitive seizures and rank the genes that are associated with the strongest light sensitivity, we have performed a multiparametric analysis that took into account seven key variables related to light-induced fish movement. Our approach allowed us to segregate crispants for the genes of interest and their scrambled counterparts in activity zones associated with low or high motor behavior. We speculate that the most extreme behaviors observed in our assay represented aberrant body movement corresponding to motor manifestations of light-induced seizures (e.g., muscular spasms). Importantly, not all crispants for a given gene displayed a strong hyperactive response to the light stimuli, while few control crispants showed a hyperactive behavior in response to the flashes of light. We believe that our analysis provides a meaningful representation of the complex etiology and manifestation of epilepsy in the population. On the one hand, mutations in epilepsy-associated genes might have a variable expressivity and an incomplete penetrance [68], making the severity of the resulting phenotypes extremely variable, as observed in our experiment. On the other hand, control crispants showing a strong light-induced motor response might be representative of cases of idiopathic photosensitivity [69]. Differently from the analysis of discrete phenotypes as the total distance moved or the maximum velocity of the larvae during the assay, we believe that our PCA and behavioral differentiation based on Mahalanobis distance allows a finer characterization of the mutant genotype/phenotype fingerprint and enables to better infer phenotype severity based on gene impairment. In our view, this represents a step towards better translatability of the results obtained with zebrafish larvae to human scenarios. Along these lines, among the genes tested, GABRA1 and SCN1A have the strongest association with photosensitivity in humans. Indeed, patients carrying mutations in one of these genes often experience seizures triggered by flashing or flickering lights [34,64]. Interestingly, the two zebrafish orthologues of these genes were identified as the highest risk factors for photosensitive seizures in our multiparametric analysis.

### 3.4. A High-Throughput Platform for ASM Drug Screening

To date, no cure has been identified for epilepsy, and the majority of medication-based approaches focus more on managing seizures or other symptoms than on reversing the underlying condition. The challenge of finding novel ASMs relies in part on the fact that an elevated percentage of patients with various kinds of epilepsy develop resistance to the treatments. Additionally, significant problems slowing down the development of novel ASMs are (1) the translatability of results obtained in studies with rodent or alternative models (human-derived iPS cells, organoids, zebrafish embryos); (2) the long time required to perform individualized compound screenings targeted at patients carrying a different mutation at the same locus or mutations in different loci.

In this study, we demonstrated that *scn1lab* crispants display a reproducible and clearly detectable epileptic phenotype, a feature that makes these models an ideal substrate for the screening of ASMs. Indeed, the possibility of relying on a behavioral readout to test for the presence of epileptic-like seizures makes it possible to test ASMs, potentially reversing the mutation-induced hyperactivity with a minimally invasive procedure. The fact that multiple larvae (up to 96) can be exposed to the epileptogenic stimulus simultaneously ensures the possibility of testing a panel of multiple ASMs in the same assay. Similarly, different concentrations of the same ASMs or different combinations of ASMs can be evaluated in a single experiment. Moreover, multiple behavioral recordings can be performed in a single experimental day, increasing further the throughput of crispants-based screenings of ASMs.

In this report, we proved that the acute incubation of *scn1lab* crispants with previously described ASMs (valproic acid and fenfluramine) is sufficient to prevent light-induced epileptic events. We believe that the chronic exposure of *scn1lab* F0 mutants to neuroactive compounds can be employed to screen for neuroprotective molecules potentially acting on the molecular mechanisms at the basis of the neural circuits’ dysfunction causing the manifestation of epilepsy. Thus, zebrafish larvae could be employed to identify novel therapeutic agents acting on the cause of the disorder and not only on its symptoms.

This strategy is not only interesting for testing novel ASMs or performing drug repurposing but also for studying the contribution of genetic modifiers, which are mutated genes that interact with the primary mutations and are able to modulate the resulting phenotypes. This is the case for the cholesterol 24-hydroxylase (CH24H) gene, which, when pharmacologically inhibited, ameliorates the phenotype and reduces the occurrence of seizures in Scn1a Dravet mouse models [70]. Since the *scn1lab* mutant is the gold standard epilepsy model in the zebrafish community, and *scn1lab* has shown a strong epileptic phenotype in our study, genetic modifiers could be identified and tested through our approach in this background, advancing the discovery of potential therapeutic targets.

Taken together, the strategy proposed here offers a high-throughput platform for drug screenings, repurposing of ASMs or tests of genetic modifiers. We demonstrate that the behavioral multiparametric analysis proposed in this study enables the evaluation of molecule efficacy, making it possible to identify compounds that inhibit the manifestation of seizures. 

### 3.5. Concluding Remarks

In this report, we have developed a strategy to expand the utility of screenings based on zebrafish F0 knock-out larvae to partially overcome the time and throughput issues in NDD functional genomics and drug testing.

Our approach allowed us to investigate the morphological and behavioral alterations induced by the loss of function of genes associated with NDDs, with a special focus on genes involved in the pathogenesis of childhood epilepsy. Moreover, we established a novel approach to employ kinematic parameters to define and identify larvae affected by light-induced seizures. This strategy allowed us to rank different candidate genes according to the severity of their phenotype and, even, to test the efficacy of different anti-epileptic compounds in F0 larvae.

All in all, we believe that the in vivo combination of F0 gene targeting and phenotypic fingerprinting of zebrafish larvae represents a powerful tool for functional genomics, drug screening and personalized medicine development that can be scalable from epilepsy to other neurodevelopmental disorders and, in principle, to a vast array of indications.

## 4. Materials and Methods

### 4.1. Zebrafish Husbandry and Breeding

Adult wild-type AB strain zebrafish were maintained at 28.5 ± 1 °C in a 14 h light/10 h dark cycle in a closed recirculating tank system, according to standard protocols [71]. Zebrafish embryos were obtained by mating wild-type adult zebrafish through standard methods. All the experiments were performed on 120 h-post-fertilization (hpf) larvae.

### 4.2. Generation of Crispants

For the generation of crispants, zebrafish embryos were injected at the one-cell stage with the corresponding CRISPR/Cas9 machinery mix. For each target gene, 2 sgRNAs were employed in order to maximize the mutagenesis rate (Table 4). As a negative control for the experiments, commercial scrambled sgRNA (5′-G*C*A*CUACCAGAGCUAACUCA-3′) was also injected in one-cell-stage embryos at the same time as the other crispants. The scrambled sgRNA binds the Cas9 but it does not target any sequence of the zebrafish genome, simulating the injection conditions in the embryo but not generating any indel in its DNA. When stated, *tyrosinase* sgRNA (5′-GGACTGGAGGACTTCTGGGG-3′) was co-injected with the sgRNAs for each target gene.

### 4.3. The Rate of Mutations Analysis

For the evaluation of the mutagenesis ratio for each childhood epileptic crispant and to assess the efficacy of co-injecting sgRNAs for our target genes and *tyr* sgRNA, 5 dpf pigmented (low *tyrosinase* efficacy) and non-pigmented (high *tyrosinase* efficacy) larvae were selected and individualized; their genomic DNA was extracted and the locus targeted by the sgRNAs of our target genes was Sanger-sequenced. With the obtained individualized sequences, the mutation ratio was quantified using the Synthego ICE analysis tool.

### 4.4. Morphological Analysis

To assess any morphological alterations in the different crispants studied, vertebrate automated screening technology (VAST Biometrica System) was used. Each larva was analyzed at 120 hpf for different morphological parameters, such as body deformity, heart edema, yolk edema, scoliosis, heart area, eye diameter, lateral and dorsal length and absence of the fin. Phenotypes were assessed using an ad hoc ImageJ/FIJI plug-in for the analysis of the morphological alteration [30]. The experimental design used a 96-well plate (131012C, Clearline) with one larva per well; a negative control group (injected with a scramble sequence) was included for each plate imaged with the VAST. Plate design for the different childhood epilepsy crispants was as follows:Plate A: scrambled, *scn1lab* and *ube3a*Plate B: scrambled, *pcdh19* and *gabra1*Plate C: scrambled, *kcnq2a*, *kcnq2b* and *kcnq2ab*Plate D: scrambled and *adgrg1*

Each plate contained a scrambled control from the same batch of embryos, injected and manipulated together with the studied childhood epilepsy mutants to minimize the possible batch effect. Statistical analysis was performed using one-way ANOVA, whereas multiple comparison analysis was performed using Dunnett’s test. The significance was considered at *p*-value < 0.05.

### 4.5. Behavioral Experiments

The behavior was analyzed using a series of different standardized paradigms. The performance of the injected larvae was recorded at 120 hpf using the EthoVision XT 12.0.30 software along with the DanioVision chamber (Noldus Information Technologies, Wageningen, The Netherlands). This system consists of a closed circulating water system maintained at 28 °C with a temperature sensor. The chamber has a camera placed above, allowing the recording of the zebrafish larvae. Adapting to the zebrafish cycle, experiments were always performed between 12.00 h and 16.00 h [72]. As stated with the morphological analysis, different distributions for the plates were used in the different behavioral analyses of the epilepsy-associated genes, always with a scrambled control in each plate to minimize the batch effect and to always be compared with.

A polarized corrector lens was used to adjust and normalize the image of the plate, allowing the recording of the whole plate equally. The threshold detection settings were optimized for each experiment to identify the larvae correctly.

#### 4.5.1. Locomotion Activity Assessment

To evaluate possible locomotion alteration in the generated crispants, the standardized 25 min dark/light cycles paradigm (5 min darkness—5 min light up to 25 min) was used. The total distance moved (mm) was analyzed. For these experiments, 48-well plates (Nunc 055431/150687, Thermo Scientific, Waltham, MA, USA) were used and placed in the chamber in line with the camera (one larva per well). To allow the habituation of the larvae to the plate, they were plated at least 3 h before the behavioral recording and all the tests had a 10 min light acclimatation phase inside the recording chamber. As aforementioned, different plate distributions were used, allowing to always have a scrambled control on each plate.

Plate A: scrambled, *scn1lab* and *ube3a*Plate B: scrambled, *pcdh19* and *gabra1*Plate C: scrambled, *kcnq2a*, *kcnq2b* and *kcnq2ab*Plate D: scrambled and *adgrg1*

#### 4.5.2. Light-Induced Epilepsy Behavior

The paradigm used in this set of experiments constituted a set of 5 flashes (1 s of light flash—49 s of darkness, repeated 5 times), performed in order to trigger epileptic behavior. For these experiments, 48-well plates (Nunc 055431/150687, Thermo Scientific, Waltham, MA, USA) were also used and placed in the chamber in line with the camera (one larva per well). To allow the habituation of the larvae to the plate, they were plated at least 3 h before the behavioral recording and all the tests had a 10 min light acclimatation phase inside the recording chamber. The same plate distributions as in the locomotion activity assay of the selected mutants were used.

Plate A: scrambled, *scn1lab* and *ube3a*Plate B: scrambled, *pcdh19* and *gabra1*Plate C: scrambled, *kcnq2a*, *kcnq2b* and *kcnq2ab*Plate D: scrambled and *adgrg1*

#### 4.5.3. Pharmacological-Induced Epilepsy Behavior

To perform the experiments, PTZ (Cas#: 54-95-5, Sigma Aldrich—Merck, St. Louis, MO, USA) was dissolved in DMSO (dimethylsulfoxide, Cas#: 67-68-5, MRI Global, Kansas City, MO, USA) to 1 M stock solutions and stored at −20 °C until needed; treatment solutions were freshly prepared before each experiment.

Given PTZ characteristics [25,28,39,40,41], the exposure time should be limited to avoid toxicity and larvae mortality. For this reason, the larvae were exposed to an acute treatment of 180′ before the behavioral assay in all the experiments. Two different concentrations, 1 mM as a sub-optimal non-epileptic concentration and 3 mM as a pro-convulsant concentration [25,42], were used in order to evaluate the sensitivity of the different studied crispants to their effects. The experiments were performed using 96-square-well plates (7701-1651, Cytiva, Washington D.C., USA). The paradigm used in this set of experiments constituted one approach to evaluate locomotor alterations and spontaneous seizures during a continuous light period. The approach consisted of 15 min of continuous light. To allow the habituation of the larvae to the plate, they were plated at least 3 h before the behavioral recording and all the tests had a 10 min dark acclimatation phase inside the recording chamber.

Analogously with the previous behavioral experiments, three different distributions of the plates were used in the behavioral analysis of the epilepsy-associated genes. 

Plate A: scrambled, *adgrg1, pcdh19* and *ube3a*Plate B: scrambled, *scn1lab* and *gabra1*Plate C: scrambled, *kcnq2a*, *kcnq2b* and *kcnq2ab*

### 4.6. Behavioral Analysis

For the statistical analysis, one-way ANOVA was used to compare each crispant with the negative control on their respective plates.

In light-induced behavior experiments, locomotion during the dark and light periods was evaluated, and the total distance moved (mm), both cumulative along the experiment and per minute, was analyzed. To describe the presence of an epileptic crisis, the maximum speed (mm/s) reached by the larvae and the number of turns performed in the 2 s following a light flash were analyzed. In order to obtain a complete picture of the behavior of the studied crispants, five different light flashes were performed, distanced one minute from each other, to perform five different triggers (or replicates) per experiment. For the final analysis, the mean of the responses of the five different flashes were evaluated.

In pharmacological epilepsy behavior experiments, the maximum speed (mm/s) reached by the larvae during the experiment was evaluated, analyzing the presence of spontaneous epileptic events along the recorded time. 

Statistical significance was evaluated using one-way ANOVA comparing the different crispants with the negative control on their plate, or a *t*-test if the comparisons were only between two conditions. The significance was considered at *p*-value < 0.05.

### 4.7. Principal Component Analysis and Mahalanobis Distance Calculations

Multivariate data analysis (principal component analysis, PCA-2D) was used to assess differences in behavioral analysis among all observations. Kinematic variables were collected from the performed behavioral experiments, after the flash-light induction, flash variables such as maximum acceleration, maximum velocity, highly mobile, mobile, mobility body percentage, turn angle number and immobile values for each larva were analyzed. PCA shows linear combinations of the variables that maximally explain the total variance of the dataset. Only variables with a contribution above a certain threshold (squared distance of the observation to the origin, cos2 > 0.75) were considered. For comparison purposes, quantile transformation was applied to filter observations below a kinematic threshold (‘colMoving’). Batch effect was tested for experiment date, replicates, flash number and other technical differences, showing no significant differences.

To measure differences in flash behavior activity, we applied a statistical test of variance called Mahalanobis distance (MD) on the PC plot. MD measures how distant a point is from the center of a multivariate normal distribution. Mahalanobis distances can be converted into probabilities using a chi-squared distribution and a significance level might be specified [73]. It is commonly applied in multivariate anomaly detection by defining two parameters, a distance magnitude (MD) and a *p*-value as a statistical measure to validate a hypothesis (outliers, *p*-value < 0.05). Using a histogram of the MD values (density plot) we detected a threshold defined between two distributions of main variances, with an MD > 2 and *p* value < 0.15 for the samples with the highest variance. A binomial test showed a high statistical difference between the number of crispants in the high variance region compared to scrambles (the proportion of crispants is 0.063 greater than expected for scrambles 0.04; *p* value < 0.001).

The R v4.2 software was used for statistical computing and graphics.

### 4.8. Pharmacology with Antiepileptic Compounds

The different compounds (topiramate (Cas#: 97240-79-4, Sigma Aldrich—Merck), valproic acid (Cas#: 1069-66-5, Sigma Aldrich—Merck) and fenfluramine (Cas#: 458-24-2, Cymit quimica, Barcelona, Spain)) were dissolved in DMSO (dimethylsulfoxide, Cas#:67-68-5, MRI Global) to 100 mM stock solutions and stored at −20 °C until used; treatment solutions were freshly prepared before each experiment.

Some of the tested drugs, such as topiramate and valproic acid, are known to have potential teratogenic and toxic effects [74,75,76], so the exposure time to these compounds should be limited. Different publications using the abovementioned compounds show different concentrations and incubation times [4,28,63,77,78]. The compounds were tested only in some of the epileptic childhood crispants that showed an epileptic phenotype.

The different doses and treatment times used were as shown below.

-Topiramate: 50 µM and 100 µM, treatment 180′ before the behavioral assay.-Valproic acid: 50 µM and 100 µM, treatment 180′ before the behavioral assay.-Fenfluramine: 17.5 µM and 35 µM, treatment 24 h before the behavioral assay.

### 4.9. The Statistics of the Pharmacology

In order to compare light flash kinematics between crispants and treated-crispant larvae samples, binomial or Fisher’s exact (smallest sample sizes) tests for binary sampling statistics were applied. The highest variance compared to low variance sample proportions based on Mahalanobis metrics was assessed for each dose and treatment between treated crispants and crispants observations. Crispants were also tested against scrambled larvae as a reference.

## Figures and Tables

**Figure 1 ijms-25-02991-f001:**
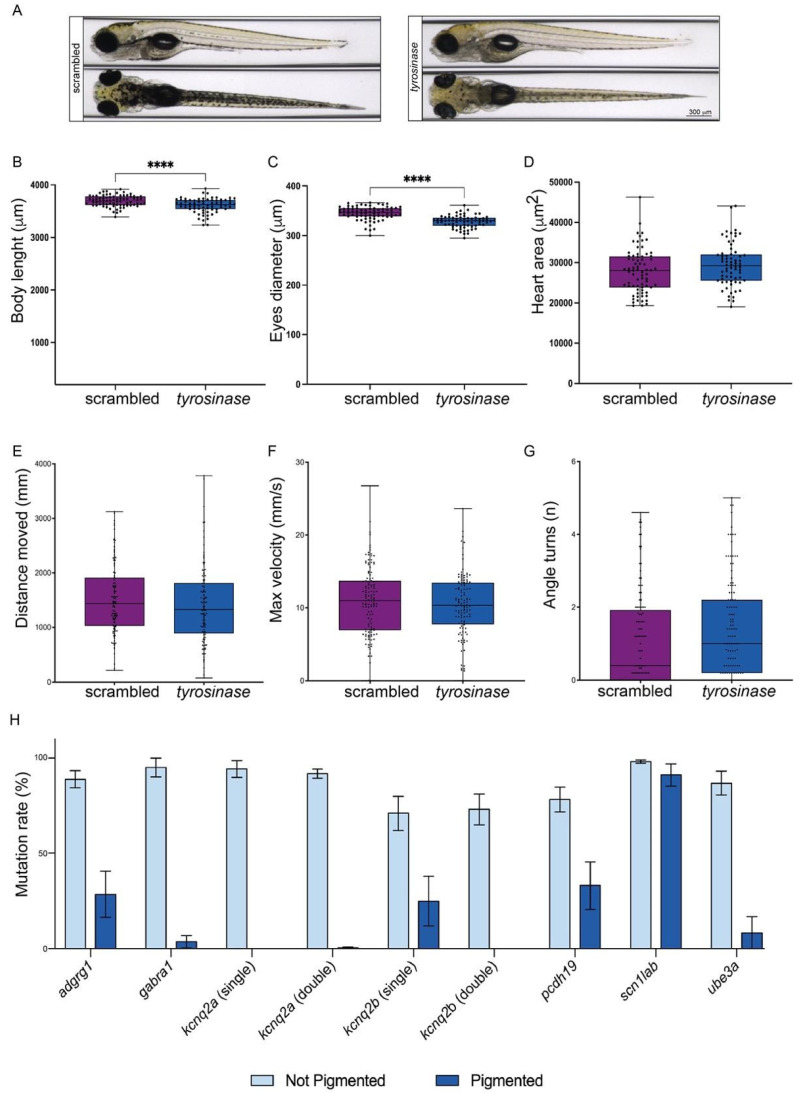
*Tyr* loss of function enables the selection of crispants with the highest mutation rate while keeping other behavioral and developmental parameters unchanged. (**A**): Dorsal (lower panels) and lateral (upper panels) images of scrambled (left panels) and *tyrosinase* crispants (right panels), with a reduction in pigmentation in the latest. (**B**–**D**): Analysis and comparison of different relevant morphological phenotypes in both crispants (scrambled are represented in purple; *tyrosinase* crispants are represented in blue. The error bar represents the minimum to maximum values): body length (µm) (**B**), the diameter of the eyes (µm) (**C**) and heart area (µm^2^) (**D**). (**E**–**G**): Analysis and comparison of the most relevant parameters related to epilepsy in both crispants (scrambled are represented in purple; *tyrosinase* crispants are represented in blue. Error bar represents the minimum to maximum values): distance moved (mm) during the dark/light cycles phase (**E**), maximum velocity achieved after the light flashes (mm/s) (**F**) and the number of angle turns after the light flashes (**G**). (**H**): Bar plot showing the mutagenesis efficiency observed in the targeted *loci* in pigmented larvae (dark blue) and unpigmented larvae (light blue). From left to right: mutation rate observed in the *adrgr1* CDS in *adrgr1* crispants; mutation rate observed in the *gabra1* CDS in *gabra1* crispants; mutation rate observed in the *kcnq2a* CDS in *kcnq2a* single crispants; mutation rate observed in the *kcnq2a* CDS in *kcnq2a-kcnq2b* double crispants; mutation rate observed in the *kcnq2b* CDS in *kcnq2b* single crispants; mutation rate observed in the *kcnq2b* CDS in *kcnq2a-kcnq2b* double crispants; mutation rate observed in the *pcdh19* CDS in *pcdh19* crispants; mutation rate observed in the *scn1lab* CDS in *scn1lab* crispants; mutation rate observed in the *ube3a* CDS in *ube3a* crispants. Error bars represent the standard error of the mean (SEM), **** *p* < 0.001 (*t*-test).

**Figure 2 ijms-25-02991-f002:**
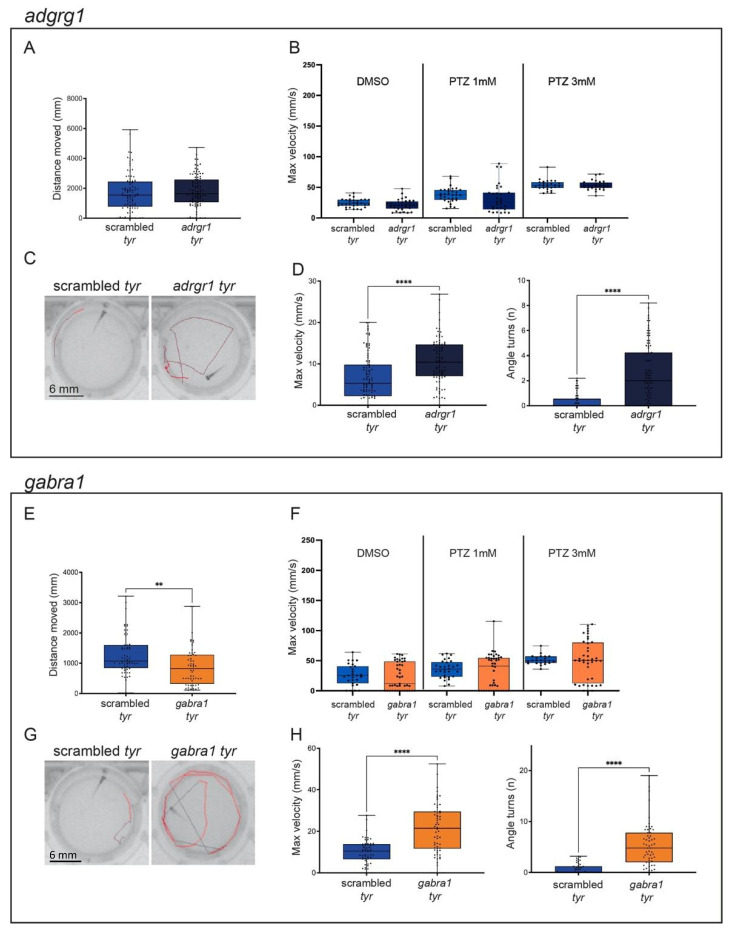
Characterization of *adgrg1* and *gabra1* crispants. (**A**–**D**). Representation of *adgrg1* measurements in behavioral studies: distance moved (mm) during the dark/light cycles (**A**), the maximum velocity (mm/s) during the light phase in response to different PTZ concentrations (DMSO, PTZ 1 mM and PTZ 3 mM) (**B**), representative tracking images of scrambled (left) and *adgrg1* crispants (right) after the light stimuli to induce convulsions (**C**) and the maximum speed (mm/s) and angle turns measures after the light stimuli (**D**). (**E**–**H**). Representation of *gabra1* measurements in behavioral studies: distance moved (mm) during the dark/light cycles (**E**), the maximum velocity (mm/s) during the light phase in response to different PTZ concentrations (DMSO, PTZ 1 mM and PTZ 3 mM) (**F**), representative tracking images of scrambled (left) and *gabra1* crispants (right) after the light stimuli to induce convulsions (**G**) and the maximum speed (mm/s) and angle turns measures after the light stimuli (**H**). Scrambled are represented in blue, *adgrg1* crispants are represented in dark blue and *gabra1* crispants are represented in orange. The error bar represents the minimum to maximum values. ** *p* < 0.01, **** *p* < 0.001 (*t*-test). Next, we decided to test the possibility of inducing epileptic-like behavior by exposing the crispants to different epileptogenic stimuli. In order to trigger seizures, we employed two kinds of stimuli, namely incubation with Pentylenetetrazole (PTZ) and exposure to intermittent flashes of light. PTZ is a pro-convulsant commonly used as a pharmacological epilepsy model [25,28,39,40,41] while flashing or flickering lights are known to cause photosensitive seizures in a large number of patients affected by epilepsy [20].

**Figure 3 ijms-25-02991-f003:**
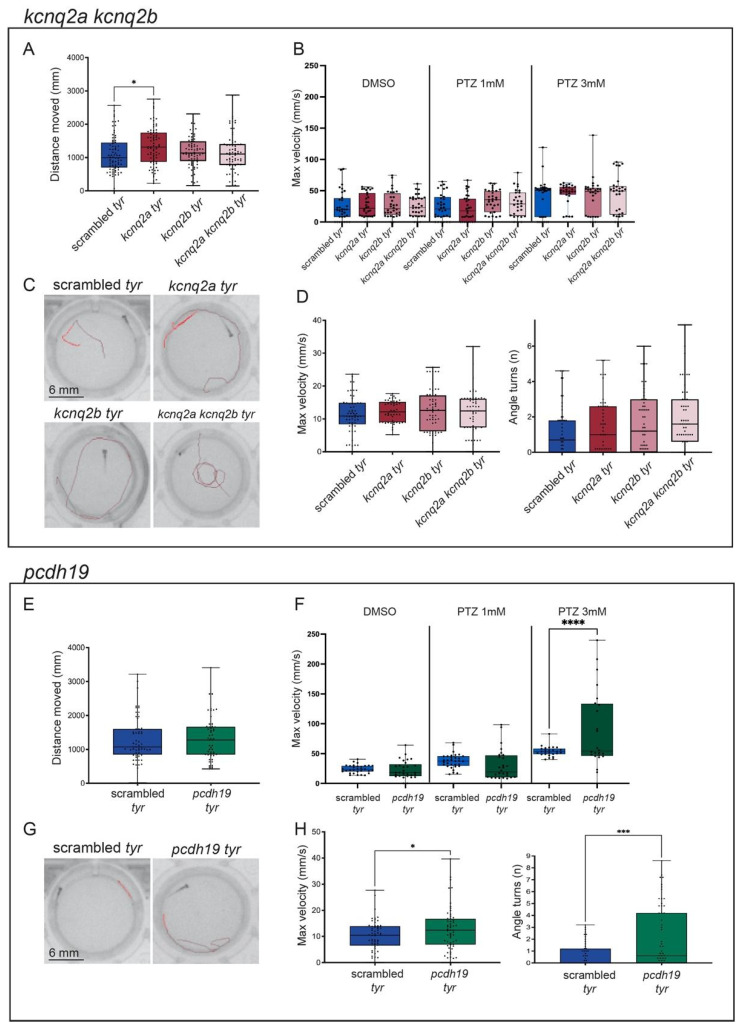
Characterization of *kcnq2a, kcnq2b*, double *kcnq2a/kcnq2b* and *pcdh19* crispants. (**A**–**D**). Representation of *kcnq2a*, *kcnq2b* and double *kcnq2a/kcnq2b* measurements in behavioral studies: distance moved (mm) during the dark/light cycles (**A**), the maximum velocity (mm/s) during the light phase in response to different PTZ concentrations (DMSO, PTZ 1 mM and PTZ 3 mM) (**B**), representative tracking images of scrambled (top left), *kcnq2a* crispants (top right), *kcnq2b* crispants (bottom left) and double *kcnq2a/kcnq2b* crispants (bottom right) after the light stimuli to induce convulsions (**C**), and the maximum speed (mm/s) and angle turns measures after the light stimuli (**D**). (**E**–**H**). Representation of *pcdh19* measurements in behavioral studies: distance moved (mm) during the dark/light cycles (**E**), the maximum velocity (mm/s) during the light phase in response to different PTZ concentrations (DMSO, PTZ 1 mM and PTZ 3 mM) (**F**), representative tracking images of scrambled (left) and *pcdh19* crispants (right) after the light stimuli to induce convulsions (**G**) and the maximum speed (mm/s) and angle turns measures after the light stimuli (**H**). Scrambled are represented in blue, *kcnq2a* crispants are represented in red, *kcnq2b* crispants are represented in pink, double *kcnq2a/kcnq2b* crispants are represented in light pink and *pcdh19* crispants are represented in orange. The error bar represents the minimum to maximum values. * *p* < 0.05, *** *p* < 0.005, **** *p* < 0.001 (*t*-test).

**Figure 4 ijms-25-02991-f004:**
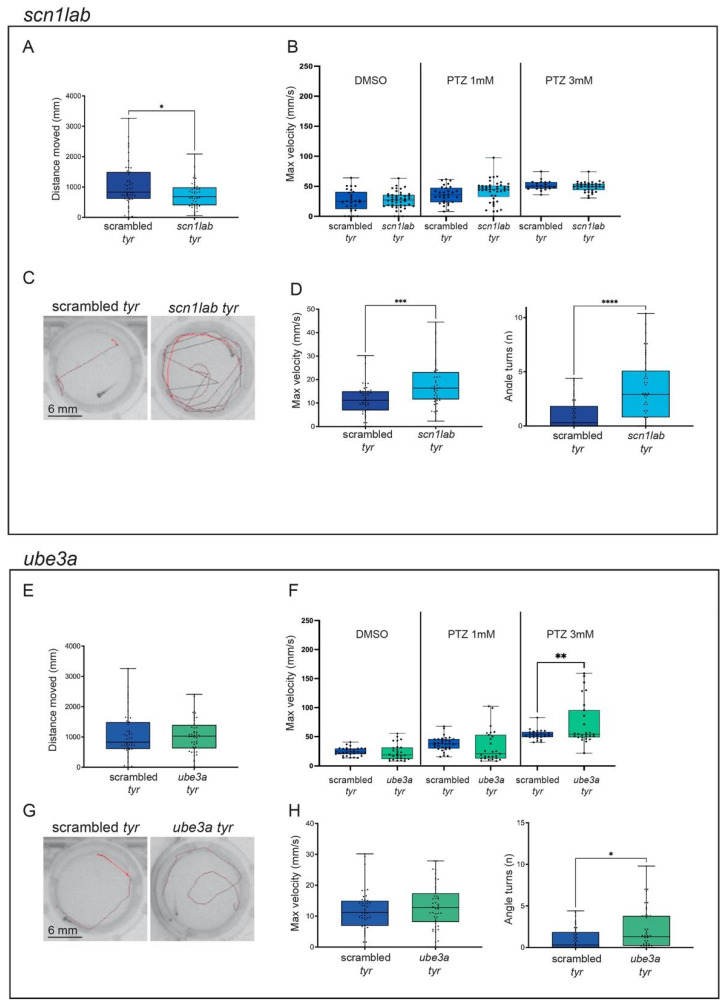
Characterization of *scn1lab* and *ube3a* crispants. (**A**–**D**). Representation of *scn1lab* measurements in behavioral studies: distance moved (mm) during the dark/light cycles (**A**), the maximum velocity (mm/s) during the light phase in response to different PTZ concentrations (DMSO, PTZ 1 mM and PTZ 3 mM) (**B**), representative tracking images of scrambled (left) and *scn1lab* crispants (right) after the light stimuli to induce convulsions (**C**) and the maximum speed (mm/s) and angle turns measures after the light stimuli (**D**). (**E**–**H**). Representation of *ube3a* measurements in behavioral studies: distance moved (mm) during the dark/light cycles (**E**), the maximum velocity (mm/s) during the light phase in response to different PTZ concentrations (DMSO, PTZ 1 mM and PTZ 3 mM) (**F**), representative tracking images of scrambled (left) and *ube3a* crispants (right) after the light stimuli to induce convulsions (**G**) and the maximum speed (mm/s) and angle turns measures after the light stimuli (**H**). Scrambled are represented in blue, *scn1lab* crispants are represented in light blue and *ube3a* crispants are represented in green. The error bar represents the minimum to maximum values. * *p* < 0.05, ** *p* < 0.01, *** *p* < 0.005, **** *p* < 0.001 (*t*-test).

**Figure 5 ijms-25-02991-f005:**
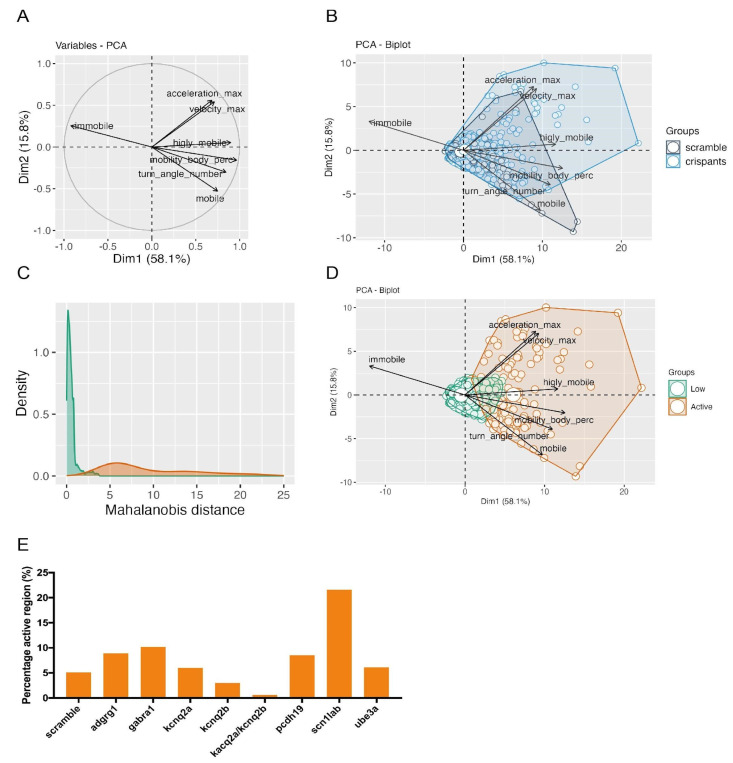
The identification of crispants with the greatest photosensitivity can be obtained through a multiparametric analysis of their reaction to light flashes. (**A**): Bidimensional plot of the different variables selected as more relevant to perform the principal component analysis (PCA), with a vectorial representation. The selected variables are maximum velocity (mm/s), the number of angle turns, maximum acceleration (mm/s^2^), angular velocity (deg/s), mobility in the arena (%) and the cumulative duration (s) of three different mobility states, immobile, mobile and highly mobile. (**B**): PCA biplot comparing two main groups, the scrambled (in black) and the different studied crispants related to childhood epileptic genes (in blue). (**C**): The definition of two different populations depending on their activity through the calculation of Mahalanobis distance. The low activity group (in green) represents the population of analyzed larvae with a non-epileptic behavior; the high activity group (in orange) represents the population of analyzed larvae with an epileptic-like behavior in response to light flashes. (**D**): PCA biplot comparing the two main groups described through the Mahalanobis distance analysis, considering the different PCA variables. (**E**): A representation of the percentage of larvae classified in the epileptic-like population of the scrambled and all the selected crispants.

**Figure 6 ijms-25-02991-f006:**
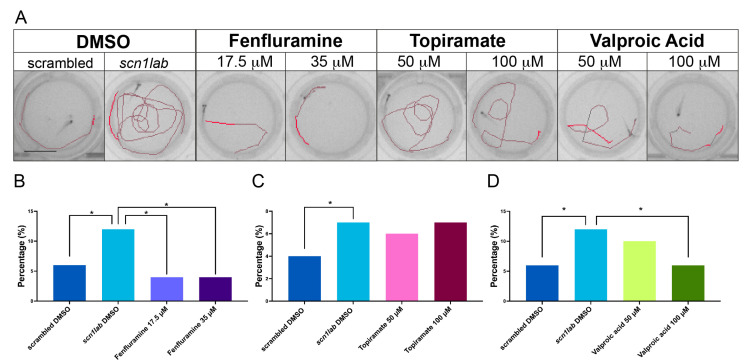
Incubating *scn1lab* and *gabra1* crispants with antiepileptic compounds offers protection against seizure-like events. (**A**): Tracking plots of the different crispants 2 s after the light flashes. Scale bar = 6 mm. The different plots correspond (from left to right) to scrambled, *scn1lab* crispants treated with DMSO (vehicle), *scn1lab* crispants treated with two different concentrations of fenfluramine (17.5 µM and 35 µM), topiramate (50 µM and 100 µM) and valproic acid (50 µM and 100 µM). (**B**–**D**): Representation of the percentage of larvae classified in the high-activity region. Scrambled treated with DMSO are represented in dark blue in all plots, *scn1lab* crispants treated with DMSO are represented in light blue in all plots. (**B**): *scn1lab* crispants treated with fenfluramine 17.5 µM are represented in light purple; *scn1lab* crispants treated with fenfluramine 25 µM are represented in dark purple. (**C**). *scn1lab* crispants treated with topiramate 50 µM are represented in light pink; *scn1lab* crispants treated with topiramate 100 µM are represented in dark pink. *scn1lab* crispants treated with valproic acid 50 µM are represented in light green; *scn1lab* crispants treated with valproic acid 100 µM are represented in dark green. In all bar graphs, * *p* < 0.05 (binomial test).

**Table 1 ijms-25-02991-t001:** Qualitative phenotypes analyzed.

	Scrambled			*Tyrosinase*		
	Number of Positive	Total Number	% of Positive	Number of Positive	Total Number	% of Positive
Body curvature	0	75	0	1	68	0.01
Snout jaw defects	2	75	0.03	0	68	0
Yolk edema	2	75	0.03	0	68	0
Necrosis	1	75	0.01	2	68	0.03
Tail bending	0	75	0	1	68	0.01
Notochord defects	0	75	0	0	68	0
Craniofacial edema	2	75	0.03	0	68	0
Fin absence	0	75	0	0	68	0
Scoliosis	0	75	0	0	68	0

**Table 2 ijms-25-02991-t002:** Candidate epilepsy-associated genes analyzed in this study.

Human Gene	Disease	Zebrafish Orthologue
ADGRG1	Bilateral frontoparietal polymicrogyria [33]	*adgrg1*
GABRA1	Different epileptic disorders [34]	*gabra1*
KCNQ2	Benign familial neonatal seizures [35]	*kcnq2a*; *kcnq2b*
PCDH19	PCDH19 Epilepsy [36]	*pcdh19*
SCN1A	Dravet syndrome [37]	*scn1lab*
UBE3A	Angelman syndrome [38]	*ube3a*

**Table 3 ijms-25-02991-t003:** The proportion of active larvae in the different crispants.

Target Gene	Number of Larvae	Number of Flash Responses	Percentage of High-Activity Responses
scrambled	116	617	5.1%
*adgrg1*	86	282	8.9%
*gabra1*	56	212	10.2%
*kcnq2a*	57	219	6%
*kcnq2b*	55	181	3%
*kcnq2a/kcnq2b*	53	208	0.6%
*pcdh19*	50	157	8.5%
*scn1lab*	46	183	21.6%
*ube3a*	44	150	6.1%

**Table 4 ijms-25-02991-t004:** Sequences of sgRNA and the corresponding primers for indels analysis and sequencing.

Zebrafish Gene	Primers	sgRNAs	Targeted Exon
*adgrg1*	FW—5′-GTCATTTCTGTGTGTTCTGGGAG-3′	5′-CGGTGCAGCAGGTTCCTTGA-3′;	Exon 2
	RV—5′-GGTGATGTTGTGATGCATGGTA-3′	5′-GTCAAAGGTGATATCATCAC-3′	
*gabra1*	FW—5′-TATTCCTTTGCACTGGCTGAGA-3′	5′-CTGCCTGAAGAACACATCTA-3′	Exon 5
	RV—5′-CGAACACAGACACCAACGAAAT-3′	5′-CCCGACACGTTCTTCCACAA-3′	
*kcnq2a*	FW—5′-CCGCCAACGGGGAAGTTTA-3′	5′-GGTAAATGAACGCCCAGCCG-3′;	Exon 1
	RV—5′-AGTTTGAGCATTCTGGGCGG-3′	5′-TCTGGAGCGACCCCGCGGCT-3′	
*kcnq2b*	FW—5′-CCAGAACAAGTTCTCCAGGGA-3′	5′-TGCTCGCACCTGCTGTAGGG-3′;	Exon 1
	RV—5′-AATTCTGCAGGCGTCGGTAA-3′	5′-TTTCTCGGCCTGCGGGGCGG-3′	
*pcdh19*	FW—5′-GGACTGGAGTCGATGCCG-3′	5′-TAACCCGCAAATAAGGCTGT-3′;	Exon 1
	RV—5′-GTGTACCGAGACTGCGTTTCT-3′	5′-CGGTACCAGATTTCCCCTAG-3′	
*scn1lab*	FW—5′-CATCAGCTCCCAGAGTGACC-3′	5′-TGCGATGCGTTGCTCGATAG-3′;	Exon 1
	RV—5′-CCTTCAGGTAGTCCTACAGCCT-3′	5′-TCTCCGTAGATGAACGGCAG-3′	
*ube3a*	FW—5′-ATCACAATGTGTACGCCGCT-3′	5′-GTTGTGGTCGCAGTACGGTG-3′;	Exon 2
	RV—5′-GATCCACTCGAGGGCCTTTT-3′	5′-AGCTCGCTGGACTCAGGGAT-3′	

## Data Availability

The data presented in this study are available on request from the corresponding author. The data are not publicly available due to internal policy of the company.

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
