# Peer review of "A Zebrafish-Based Platform for High-Throughput Epilepsy Modeling and Drug Screening in F0"

_ijms, 2024, doi:10.3390/ijms25052991_

Round 1
Reviewer 1 Report
Comments and Suggestions for Authors
This study by Locubiche et al., introduces a high-throughput zebrafish platform for efficient F0 knock-out (KO) model generation and neuroactive compound screening. Employing kinematic parameters, the approach identifies seizure-like events, enabling early efficacy testing of compounds. This strategy holds promise for diverse applications, fostering high-throughput drug discovery and advocating zebrafish use in personalized medicine and neurotoxicity assessment.
Revision:
Figure 1H: Were crispants evaluated for mRNA or protein levels? Please provide information on which exon/domain the sgRNA targets for the different chosen genes. Kindly include details regarding this aspect.
Figure 2: In Figure 2A, the track is not representative of the results shown in panels B, E, H, and K. For example, in the case of gabra1, the distance moved is decreased after the flash, which differs from the track image where seems increased the distanced swum. The red line in the scrambled that follows a linear trajectory along the wall is not present compared to the distance moved in gabra1.
Moreover the distance moved in 2K is strongly reduced compared to 2B (adgrg1, scrambled = 1728 mm, and scn1lab: scrambled = 1062.03 mm). How can this difference in the distance moved be explained if the embryos were analyzed at the same stage and under the same protocol?
Figure 4B: Were the compounds also tested in scrambled samples? Is the graph missing dot plots and sample numbers?
Method Section: Include a subsection describing the method for evaluating the percentage of mutations.
Author Response
We would like to extend our gratitude to the reviewer for his/her appreciation of our work and their insightful comments and constructive feedback on our manuscript. Her/His thorough review process has significantly enhanced the quality and clarity of our work. We have carefully considered each of their suggestions and revisions, and as a result, we believe the manuscript has been substantially strengthened. We are grateful for the opportunity to address their concerns and incorporate her/his valuable input into this revised version. We are confident that these improvements will contribute to a more clear and useful manuscript for the scientific community working with epilepsy, zebrafish and disease modeling in general.
Please find attached a point by point answer to the comments.

Reviewer 2 Report
Comments and Suggestions for Authors
For the publication acceptance of the manuscript, this reviewer recommends going through some issues detected.
MINOR ISSUES
There are several shortcomings concerning content as well as misspellings. Listed below are selected examples:
- line 20= de facto, check through the manuscript and make it consistent;
- line 88= in response to response;
- Table 1: body curvature is reported twice;
- line 160= phenotypes misspelled;
- line 408= therapeutic misspelled;
- line 715= one way ANOVA, check through the manuscript and make it consistent;
- line 739= we applied misspelled.
MAJOR ISSUES
1. Table1: first and last lines present the same observation, one should be deleted from the table. The sample size (N=28 for scrambled; N=25 for Tyrosinase) may be too small. I would suggest the authors to increase the number of animals analyzed.
2. A specific comparison between crispants generated in the present study and the stable KO lines previously described in the literature should be added in the manuscript.
3. Since it is well known that zebrafish locomotor activity is sensitive to the time of day, all the information concerning the Zeitgeber time of the experiments must be stated.
Author Response
We would like to extend our gratitude to the reviewer for insightful comments and constructive feedback on our manuscript. Her/His thorough review process has significantly enhanced the quality and clarity of our work. We have carefully considered each of their suggestions and revisions, and as a result, we believe the manuscript has been substantially strengthened. We are grateful for the opportunity to address their concerns and incorporate her/his valuable input into this revised version. We are confident that these improvements will contribute to a more clear and useful manuscript for the scientific community working with epilepsy, zebrafish and disease modeling in general.
Please find attached a point by point answer to the comments.

Reviewer 3 Report
Comments and Suggestions for Authors
The authors present an interesting study using zebrafish larvae F0 as a screening tool for anti-epileptic meds. One-cell embryos were injected with 3 sgRNAs (2 targeting a specific gene and 1 a "reporter" gene, here identified in tyr) and experiments were performed on larvae meeting the phenotypic criteria at 120hpf. The study is presented in a clear and easy-to-understand way, but it could use few improvements. As the authors recognize in their limitations section, tyr might not be sufficient to identify larvae with higher mutagenesis (mutagenesis level also might differ and not be uniform in the larval population) and also the canonical way to simulate seizure is not used here (PTZ seizure induction). The following are some questions/comments that I would like to see addressed in the paper:
1. I am a little skeptical in using tyr as the only reference for mutagenesis level as in the past I've observed pigmentation coming back. Have the authors seen pigmentation re-expression or changes after 120hpf?
2. Mutations rate (fig 1H) for scn1lab crispants is extremely high both in pigmented and unpigmented larvae. How do the authors explain this? Considering this crispant is the one used for the drug testing, more details might be beneficial.
3. In the same figure (1H), we can see 4 crispants for the KCNQ2 gene but in the text (line 208) authors say that only 3 were generated for this gene. Please revise this figure accordingly. Please revise also in the text (lines 221-225) where 4 crispants are still listed, unless a real difference is present for the double crispant. If so, please explain.
4. At what time point DNA was extracted? What did the sequences reveal? It would be interesting to see some examples of the Sanger sequencing in the supplemental figures. Did you detect mosaicism or clear sequence alterations (i.e. homozygous deletion etc..)?
5. Considering that the canonical way to induce seizure is via PTZ, I would like to see a comparative experiment where the authors use the same F0 approach and then test PTZ instead of flashes for seizure induction. The automated pipeline could probably be adapted to automatic detection of similar movements after PTZ. If not, the 3-phase scale from Baraban et al could be applied for manual/visual seizure classification.
6. The authors tested 6 genes but only 2 were actually showing seizure-like events. Did the chosen genes have reported photosensitivity/photo-induced seizures? If not, why were these genes chosen? Please add a paragraph in the discussion explain this. In the same paragraph, it could be worth mentioning the 2/6 genes hit as a limitation as well, especially when 1 of this is the scn1lab crispant, with a very high mutagenesis rate, independent from tyr expression.
7. I would suggest that some of the conclusions will be soften, considering my comments above - ie. 562-563 or 632-636, unless the suggested experiment and sequences will be added.
8. Please in table 4 add the tyr sequence and the scrambles as well (or at least reference to which company you purchased them from with a catalog number) used in this study.
9. If possible, add from which company the drugs were obtained (company and catalog number).
10. No reference to an animal protocol were made in the paper. Is that because zebrafish experiments do not require an animal protocol in the authors' country?
Comments on the Quality of English LanguageThe language flows well but some typos and few unclear sentences were detected.
1. Line 71 - please remove "more"
2. Line 88 - there is a duplicate "response to". Please revise.
3. Line 92-93 - please add the s to ASM as it seems that it is referring to multiple medications.
4. Table 1. Please correct Dnout in Snout and remove the last line with body curvature (as it is duplicated).
5. Line 408. Please fix the typo (therapeutic).
6. Line 423. Please fix the typo (trajectory).
7. Line 445. Please remove the double stop period (.) and add one space after "plots".
8. Line 553. Control scrambled should be reported as scrambled control.
9. Line 606. Please correct "these" in "this" as model is singular.
10. Line 739-740 could reuse rewriting. There is a typo ("awe") and the sentence has two verbs.
11. Line 746 also needs rewriting as it is unclear as it is right now (different verbs tense and maybe a missing subject?).
12. Line 754 could use the addition of "which" after the comma.
Author Response

(The authors gave the same response as above.)

Round 2
Reviewer 2 Report
Comments and Suggestions for Authors
Thank you for the revisions.
Please proof-read the final version as several misspellings, double spaces and double commas are present throughout the manuscript.
Reviewer 3 Report
Comments and Suggestions for Authors
Thank you for your revisions. Please double check that supplementary figure 4 has the correct caption. Few minor edits are needed:
-line 294 please add a space to separate the figure caption from the text
-line 296 please correct to into two
-line 305 please correct or into our
-line 315 might need some editing because it is confusing with the figure citation and the brakets and commas.
-line 341 it seems there is a double comma
Please proof-read the final version as double spaces might be present throughout the paper.
Comments on the Quality of English LanguagePlease see above.